# Study of Infrared Laser Parameters on Surface Morphology and Hydrophobic Properties

**DOI:** 10.3390/ma12233860

**Published:** 2019-11-22

**Authors:** Xia Ye, Jiang Gu, Zhenmin Fan, Xiaohong Yang, Wei Xu

**Affiliations:** 1School of Mechanical Engineering, Jiangsu University of Technology, Changzhou 213016, Chinafanzhenmin2009@163.com (Z.F.);; 2Department of Materials Engineering, Jiangsu University of Technology, Changzhou 213016, China; yxh@jsut.edu.cn

**Keywords:** infrared laser, laser parameters, surface morphology, superhydrophobic

## Abstract

Many studies have shown that super hydrophobic surfaces have been applied to micro–nano structures and low surface energy materials. In the present study, infrared laser scanning and simple salinization modification were used to improve the hydrophobicity of a surface. When the scanning speed was 100 mm/s, the laser power was 30 W and the scanning interval was 200 μm, the apparent contact angle of surface was up to 157°. The assessment of surface characteristics revealed that decreasing scanning speed or increasing laser power were able to improve the hydrophobicity of the surface. After aging treatment, the superhydrophobic surface prepared by this method still had good durability.

## 1. Introduction

Superhydrophobicity has attracted the attention of many researchers due to the great importance in both its fundamental research and applications [1]. A surface with a water contact angle greater than 150° and contact angle hysteresis lower than 10° is called a superhydrophobic surface [2]. Hence, in view of the specific wetting state [3], the superhydrophobic surface can be used in many fields, such as drag reduction [4,5], antifrosting [6], corrosion resistance [7,8], self-cleaning [9,10] and microfluid transportation [11,12]. The 6061 aluminum alloy is an aluminum–magnesium–silica alloy which has excellent electrical conductivity, low-specific weight and high-specific strength. So, it is widely used in aircraft, food industry machinery, architecture and transport. Nevertheless, the surface of aluminum alloy exhibits hydrophilicity [13]. This hydrophilic peculiarity limits the application of aluminum alloy in some special fields, such as marine application [14]. Therefore, it is necessary to change the wettability of aluminum alloy surface.

Generally, there are two ways used to fabricate superhydrophobic surfaces. One is creating micro–nanometer hierarchical structures on a hydrophobic surface. The other is reducing surface free energy on the rough surface. The specific preparation methods of superhydrophobic surfaces include plasma treatment [15,16], electrochemical deposition [17], phase separation [18,19], chemical etching [20], electro-spinning [21], sol–gel processing [22] and laser irradiation [23,24,25,26,27,28,29]. Laser irradiation is able to achieve stable superhydrophobic samples. Due to the hydrophilicity of the metal, reducing surface energy of rough surfaces is widely used to achieve superhydrophobic surface on the metal. Moradi et al. [30] reported the superhydrophobicity of samples enhanced by high energy density of the femtosecond laser and relatively high scanning speed. Tang et al. [31] indicated that the wettability of these surfaces may be changed after laser marking, as microstructure greatly affects the wettability of the surface. Ta et al. [32] found that the most suitable spacing for achieving incomplete wetting surfaces in the shortest exposure time is 50–150 μm. During these intervals, stable Cassie-Baxter state superhydrophobic surfaces can be fabricated. Song et al. [33] proved that superhydrophobic surfaces could be fabricated on Al by femtosecond laser. The adhesion and superhydrophobicity of the surface can be tuned by adjusting laser processing parameters. Ahmmed et al. [34] showed that the regular hierarchical structures, such as square pillars, parallel grooves, pyramids, columns and hole structures, can be induced by femtosecond laser on the copper surface. However, some of the methods have many limitations. For example, they may be expensive, have strict requirements of conditions, or be unstable and uncontrollable [35]. In these aspects, the infrared laser irradiation is different from these other methods. By changing laser parameters, the preparation process can be controlled. Inexpensive and stable superhydrophobic surfaces can be easily fabricated [36].

In this work, we used infrared laser irradiation to fabricate superhydrophobic surfaces on Al alloy surfaces. Meanwhile, hexadecyltrimethoxysilane (H_3_C(CH_2_)_15_Si(OCH_3_)_3_) was used to modify samples for decreasing the surface energy. Laser processing parameters were changed to find out the optimal settings in order to fabricate a stable superhydrophobic surface on aluminum alloy. By changing scanning speed, laser power and scanning interval, different microstructures were created. The surface morphology was measured by using a scanning electron microscope (SEM). Also, 4-μL distilled water droplets were used to characterize the apparent contact angles of the microstructural surface. The results indicate that a large amount of air can be stored in the microgrooves when the scanning interval is 50–200 μm. With the low scanning speed and the high laser power, the roughness of microstructure gradually became larger. Finally, the stable superhydrophobic surface can be fabricated. This study provides a facile method of fabricating superhydrophobic surfaces on Al alloy materials which enhances the application field of research and industry.

## 2. Experimental

In this experiment, Al 6061 alloys with a size of 20 × 20 × 2 mm^3^ were used. The composition of Al 6061 alloys is as follows: Mg: 0.8–1.2%, Si: 0.4–0.8%, Fe: 0.7%, Cu: 0.15–0.4%, Mn: 0.15%, Cr: 0.04–0.35%, Zn: 0.25%, Ti: 0.15%. The rest is Al. The substrates were polished by using sandpapers whose grit ranged from P 400 to P 7000. Then, the samples immersed in absolute ethanol were rinsed by ultrasonic cleaning machine (KQ-300VDE, Kunshan Ultrasonic Instruments Co, Ltd., Kunshan, China).

Various 25–300 μm microstructures on Al 6061 alloy surfaces were fabricated by an infrared laser marking machine (DL-TG-IRF-30, Suzhou Delong Laser Co, Ltd., Suzhou, China). The samples were textured by Gaussian laser pulses with laser pulse repetition rate of 20 KHZ, wavelength of 1064 nm, focused beam spot size of 75 μm and pulse duration of 100 ns. The laser power *P* ranged from 10 to 30 W. The scanning speed *v* was changed from 10 to 4000 mm/s. The scanning interval was set in the range of 20 to 300 μm. The samples immersed in absolute ethanol were rinsed by ultrasonic cleaning machine to remove slag after laser irradiation.

The flat and textured samples were bubbled in 2% hexadecyltrimethoxysilane (H_3_C(CH_2_)_15_Si(OCH_3_)_3_) ethanol solution and kept at 60 °C for 2 h in a water-bath. Then, the samples were dried at 130 °C for 30 min in a constant temperature drying oven (9503, Shanghai Aozhen Instrument Manufacturing Co, Ltd., Shanghai, China). After chemical modification, the apparent contact angle of the untreated sample was 103°.

The irradiated surface morphology was characterized by using a scanning electron microscope SEM (EVOMA10, ZEISS, Oberkochen, Germany). With the contact angle measuring instrument (CA100C, Shanghai Innuo Precision Instruments Co, Ltd., Shanghai, China), 4 μL of distilled water was used to measure the apparent contact angle. For each sample, the average values were measured in at least five different positions. The five measured positions were near the four boundaries of the sample and at the center of the sample. The deviation of the average values was ±0.8°.

## 3. Results and Discussion

Figure 1 is a comparison diagram of sample morphology with different scanning spacing when the laser power is 20 W and the scanning speed is 100 mm/s. When the scanning distance is much smaller than the spot diameter, the repetition rate of the spot is high. A cotton irregular convex structure shown in Figure 1a is textured because the cross-scanning of each laser spot makes the slag accumulate with each other. With the increase of interval, the cotton structure is gradually separated and turned to rule stripe structure. However there is still much accumulated slag between two rule stripe structures in Figure 1b. When the scanning distance is larger than the spot diameter, the stripe structure is gradually separated. Furthermore, the independent microgroove structure is formed. Due to the ablation of the laser, a certain convex structure takes shape at the edge of the microgroove in Figure 1c,d.

With the changes of laser scanning speed, there are different microstructure patterns formed on the sample surfaces. Hence, the influence of laser scanning speed on the morphology of the sample is studied under the condition of a scanning distance of 100 μm. Figure 2 shows SEM of different scanning speeds at 10 W laser power. When the laser power is low, three different microstructure patterns can be formed on the surface of the sample by changing the scanning speed. As shown in Figure 2a, the surface of the sample has an obvious groove structure because the scanning speed is low. From the enlarged view, it can be seen that the convex parts adjacent to the two grooves have micro–nanoscale chapped structures. Compared to the convex parts, the bottom of the groove is smooth without obvious micro–nano structure. Figure 2b indicates that the surface of the sample forms a tile-like structure covered layers by layers, when the scanning speed is about 500 mm/s. We can see that there are microscale chapped structures between two columns of tile structures from the enlarged view. In addition, the tile-like structure is covered with slag materials with different shapes after laser etching. With the increasing of scanning speed, the two adjacent tile structures are separated. A crater-like structure is formed, as shown in Figure 2c,d. It can be observed from the enlarged view that there are microscale chapped structures between the craters, and the outer ring of the craters is provided with etched slag material.

A single pulse of the laser may create a spot on the surface of the sample. The change of scanning speed varies the number of pulses per unit area, so that the distance (*s*) between the centers of adjacent spots can be expressed as:*s* = *v*/*f*(1)
where *v* denotes scanning speed. Also, *f* expresses the laser frequency which is set to 20 KHz in this paper. It can be known from Equation (1) that the scanning speed is proportional to the spacing of the spots. When the scanning speed is low, the overlap of the spots is high. Due to the cumulative effect of multi-pulse ablation, a continuous groove structure can be fabricated on the sample surface. With the increase of scanning speed, the overlap rate of the spot decreases, and the tile-like structure is gradually formed. When the scanning speed increases to a critical value (i.e., *s* equal to the spot diameter), the spots are no longer overlapping with each other and a crater-like structure takes shape.

Figure 3 shows SEM of different scanning speeds at 20 W laser power. By the change of the scanning speed, four different microstructural patterns can be formed on the sample surface. Figure 3a shows that a continuous microgrooved structure emerges on the sample at low scanning speed. Meanwhile, a convex structure takes shape on the edge of the microgroove with continuous accumulation of remelting material. It can be seen that the microgrooves are rather deep and the convex structure between the grooves is covered with a number of granular microstructures. In Figure 3b, when the scanning speed is changed to 500 mm/s, the stacking structure no longer appears between the two grooves. As shown in the enlarged drawing, there are still smooth surfaces which are not textured by laser between two grooves. At the same time, many laser ablations are left behind the bottom of the microgrooves. Figure 3c indicates that the tile-like structure is gradually formed on the sample surface at the scanning speed of 2000 mm/s. As shown in the enlargement, the chapped structure appears between the two column tile structures which are covered by a small amount of molten material. With the increase of scanning speed, the two adjacent tile structures are gradually separated, arising from the same crater-like structure as Figure 3d. Nevertheless, the diameter of the laser is increased to 93 μm during the increase of laser energy density. It can be seen in the big picture that the interior of the crater structure is relatively smooth. At the same time, there is a molten material on the edge of the crater.

Figure 4 shows the relationship between ablation size and laser energy density. The energy density of each point is unevenly distributed in space. The intensity at the center of the spot is largest, while the intensity at the edge is gradually decreasing [37,38]. Therefore, laser processing has the ablation threshold called I_th_ (the minimum energy density required for material to be removed). When the energy density is greater than the ablation threshold, more energy is absorbed by the sample. At the same time, there is a deep hole on the sample surface. When the energy density is less than the ablation threshold, there is no obvious surface ablation phenomenon in the peripheral area of the spot. With the increase of the sample surface temperature during laser processing and the cumulative effect of energy, a large amount of heat is absorbed by the material around the spot which results in the melting, evaporating and re-melting process of the material. This can explain the generation of chapped structure, as shown in Figure 2.

Similarly, Figure 5 exhibits that four different microstructural patterns can be constructed with the change of the scanning speed. Compared to Figure 3, it is found that the increase of laser power adds the melting substance on the surface. In addition, the diameter of the crater structures is enlarged. The etching rate of the laser is related to the energy density of the mono-pulse. When the energy *E* of the mono-pulse is very great, the absorption of the material is correspondingly high. The single-pulse energy can be expressed as:*E* = *P*/*f*(2)
where *P* represents the laser power. The power of the laser is proportional to the energy of the single pulse. When the laser power is low, the energy that acts on the sample surface is equally low. Thus, the removal rate of the material is not high. Therefore, the microstructure is shallow. On the contrary, with the increase of the laser power, the energy absorbed by the material is increased. Moreover, the surface temperature rises. After a lot of ablation and accumulation, the deeper microstructure and the wider spot are formed. Finally, the roughness of the surface is increased.

Figure 6 shows the variation trend of the microstructural models with the change of the laser power and the scanning speed. From the whole diagram, we can see that the change of the laser power and the scanning speed can make four different microstructural patterns on the surface of the sample. When the laser power is low, the deeper microstructure can be obtained by decreasing the scanning speed. However, when the laser power is at 10 W, the energy density of the mono-pulse is too low to form the accumulation phenomenon of molten material on the sample surface. So, the microgroove structure with edge protruding cannot be fabricated by relatively low laser power.

Figure 7 is the apparent contact angle curve of different scanning speed and laser power. It can be seen from the diagram that the apparent contact angle decreases with the increase of scanning speed. When the laser power is 10 W, the average apparent contact angle is minimum, about 110°. Therefore, when the laser power is low, the superhydrophobic surface cannot be formed. When the scanning speed is constant, with the increase of laser power, the apparent contact angle is gradually greater than 150°. At the same time, when the laser power is constant, the apparent contact angle becomes larger with the decrease of the scanning speed. It is believed that the decrease of scanning speed and the increase of laser power makes the roughness of the microstructure larger and the depth of the groove deeper. Thus, plenty of air can be stored. Finally, superhydrophobic surface takes shape. With the increase of scanning speed and the decrease of laser power, the roughness of microstructure becomes smaller and the depth of grooves becomes shallow. Only a small amount of air can be stored, so the contact angle decreases.

Figure 8 is the curve of apparent contact angle with different scanning distances (where the laser power is 20 W and the scanning speed is 100 mm/s). When the scanning distance is 20 μm, the apparent contact angle is 134°. With the increase of the scanning distance, the apparent contact angle increases synchronously. The microstructural groove can store a large amount of air to maintain a stable Cassie state when the scanning distance is 50–200 μm. The wetting state is changed from solid–liquid two phase combination to solid–liquid–gas three phase combination. However, the apparent contact angle decreases when the scanning distance is more than 200 μm. When the scanning distance is greater than 200 μm, the number of grooves under the droplet decreases. As a result, air cannot be stored in the microgroove structures leading to the Cassie to Wenzel transition. So, the contact angle decreases gradually.

Durability has always been a bottleneck restricting the development of superhydrophobic surfaces. In order to detect the durability of superhydrophobic surfaces, we exposed the prepared superhydrophobic surface in air for 20 months. Figure 9 is the comparison of apparent contact angle between 20 months ago and now. The apparent contact angle decreases slowly with time, but the surface is still superhydrophobic. Meanwhile, the surface wetting state is still the stable Cassie state. The microstructures will not fall off since it is fabricated by laser. The modifying agent is bonded to the substrate through hydroxyl groups. Therefore, both microstructures and modifying agent can be tightly bonded to the substrate. The microstructural surface prepared by this method has good stability and durability [39].

## 4. Conclusions

We successfully constructed a durable superhydrophobic surface on aluminum alloy surfaces by using an infrared laser-textured and chemically modified method. The effects of different laser processing parameters on the microstructure morphology are studied. By changing the laser processing parameters, it had been found that the best processing parameters are 100 mm/s, 30 W and 200 μm.

It is shown in this study that the different scanning distances made the sizes of microgroove structures change constantly. The most suitable scanning distance is between 50 and 200 μm.

We found that low scanning speed increased the pulse number per unit area. A continuous and clear microgroove structure was fabricated. With the increase of scanning speed, the laser spots were gradually separated. The crater microstructure was textured on the sample surface. The surface prepared by the low scanning speed (10–500 mm/s) had good superhydrophobic properties.

It was discovered that the change of laser power had little effect on the surface morphology, but the roughness of the microstructure increased with the increase of laser power. Thus, the appropriate laser power is from 20 to 30 W.

The superhydrophobic surface prepared by infrared laser has good durability in the atmosphere. The surface can maintain stable Cassie wetting state for a long time. Moreover, the processing method is cheap and high in production efficiency. It is expected to be applied to industrial production.

## Figures and Tables

**Figure 1 materials-12-03860-f001:**
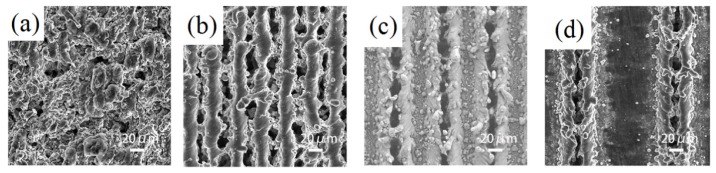
The diagram of space comparison (the power is 20 W and the scanning speed is 100 mm/s). (**a**) 20 μm; (**b**) 50 μm; (**c**) 100 μm; (**d**) 200 μm.

**Figure 2 materials-12-03860-f002:**
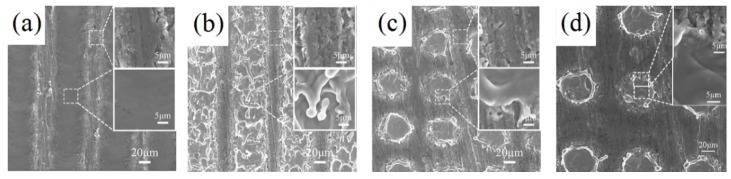
Scanning electron microscopy of different scanning speeds when the laser power is 10 W. (**a**) 10 mm/s; (**b**) 500 mm/s; (**c**) 2000 mm/s; (**d**) 4000 mm/s.

**Figure 3 materials-12-03860-f003:**
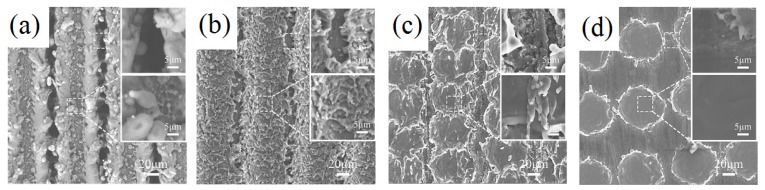
Scanning electron microscopy of different scanning speeds when the laser power is 20 W. (**a**) 10 mm/s; (**b**) 500 mm/s; (**c**) 2000 mm/s; (**d**) 4000 mm/s.

**Figure 4 materials-12-03860-f004:**
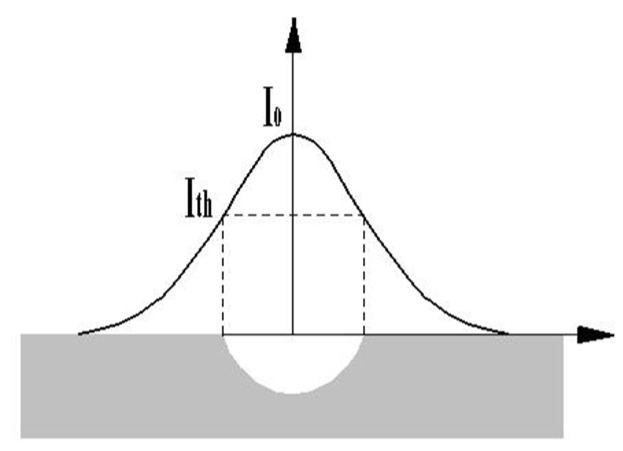
The relationship between ablation size and laser energy density.

**Figure 5 materials-12-03860-f005:**
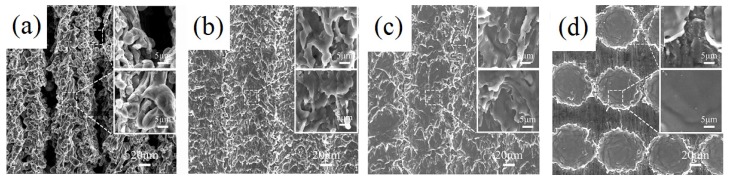
Scanning electron microscopy of different scanning speeds when the laser power is 30 W. (**a**) 10 mm/s; (**b**) 500 mm/s; (**c**) 2000 mm/s; (**d**) 4000 mm/s.

**Figure 6 materials-12-03860-f006:**
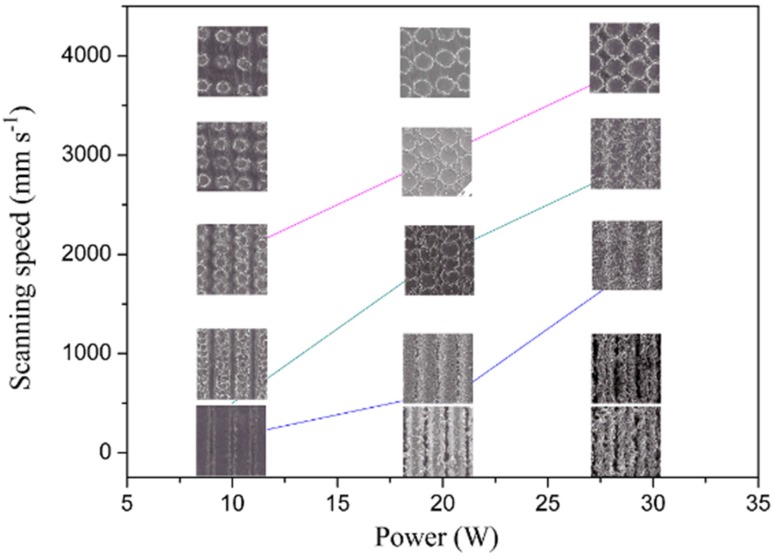
Microstructural diagram of different scanning speed and laser power.

**Figure 7 materials-12-03860-f007:**
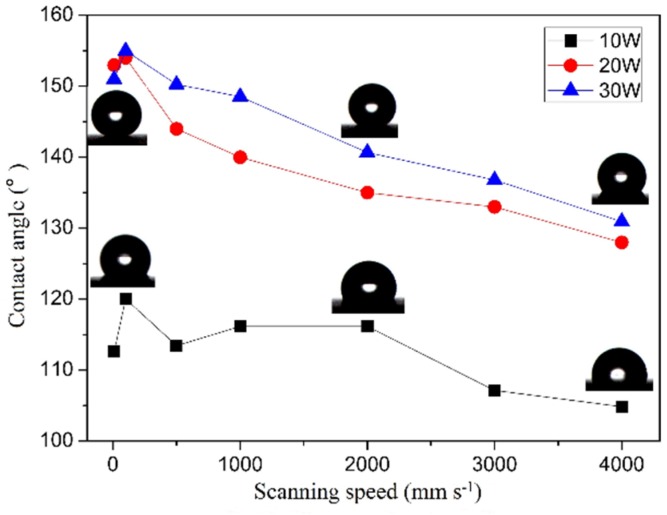
Contact angle of different scanning speed and laser power.

**Figure 8 materials-12-03860-f008:**
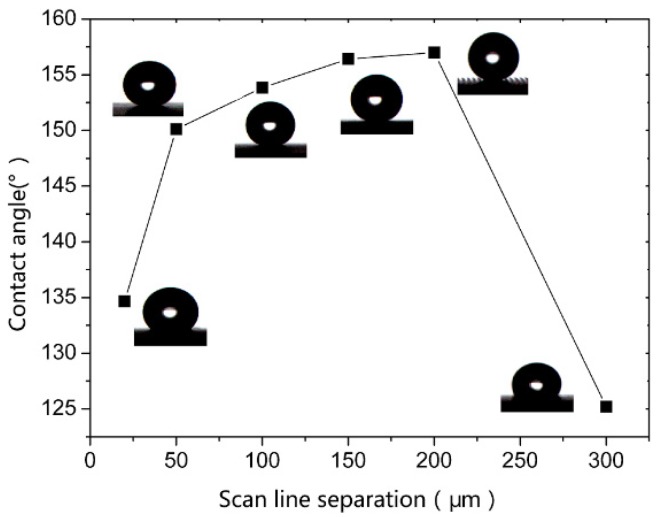
Contact angle diagrams with different scanning distance.

**Figure 9 materials-12-03860-f009:**
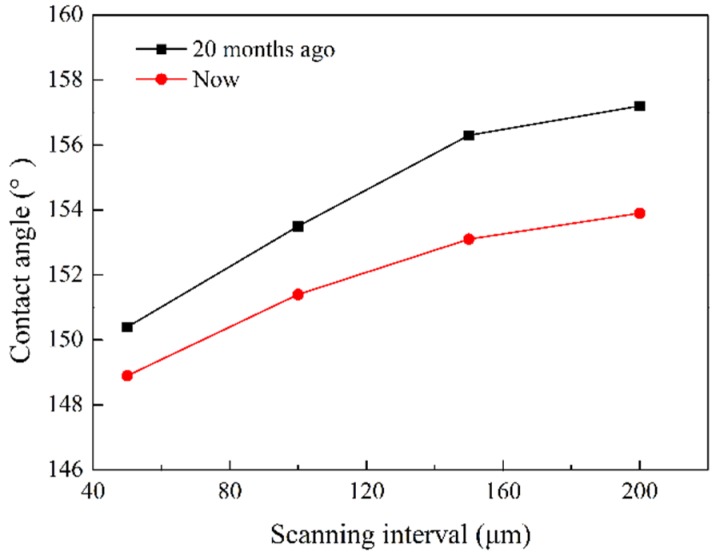
Comparison of contact angles before and after 20 months.

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
