# Peer review of "Study of Infrared Laser Parameters on Surface Morphology and Hydrophobic Properties"

_materials, 2019, doi:10.3390/ma12233860_

Round 1
Reviewer 1 Report
The manuscript present a procedure to fabricate superhydrophobic surfaces. The main advantages of the processing method are the durability of the superhydrophobic surface and the low cost and production efficiency of the method.
The characteristics of the surface treated with the proposed method are clear, but there are some points to clarify in order to describe the durability of the superhydrophobic surface and the cost and the production efficiency of the method. Specifically, a comparison between the proposed method and those methods used nowadays to fabricate superhydrofobic surfaces in terms of the durability of the superhydrophobic surface and the cost and the production efficiency of the method.
The contact angle evolution in two moments in time has been measured to study the durability of the super hydrophobic surface. Is there a linear relationship between the contact angle and the time interval between measures? The tendency of the relationship between the contact angle and the time interval between measures should be clarify in order to understand the meaning of figure 9, the sentence in line 217 (The contact angle decreases slowly with time) and the sentence in lines 220-222 (Thus, the contact angle will not rapidly decrease as time increasing. This super hydrophobic surface has good durability).
A reference should be added too or a more detailed explanation in order to explain what is considered “good durability” (line 222).
Finally, some aspects need to be detailed as weel:
In line 12, the format of the units “mm s-1” should be revised.
In line 48, and according with the previous paragraph a reference or a more detailed explanation should be added in order to clarify the sentence “Inexpensive and stable super-hydrophobic surfaces are easily fabricated”.
In line 53, the word “para-meters” should be revised.
In line 67, the number of the micro and nanometer-scale structures used in the analysis should be indicated as well as the deviation of the average values (line 80) and the description of the positions used to measure the samples (line 80). This could help to better understand the results show in figure 6.
In line 94, the format of the units “mm s-1” should be revised.
In line 100 “is10W” should be revised.
In line 107-109, figure 2(b) is associated whit a scanning speed of 1000mm/s but in line 101 Figure 2(b) indicates 500mm/s.
In line 161-166, the format of the paragraph is different from that in the rest of the paper.
Figures:
Figure 1, 2, 3, 5, the indicators (a), (b), (c) and (d) are in white inside the image and are hard to see.
The quality of Figure 9 should be improved.

Author Response
Dear reviewer:
Many thanks for the insightful comments and suggestions of you. I have made corresponding revision according to your advice. Words in red are the changes I have made in the text.
The following is the answers and revisions I have made in response to your questions and suggestions.
Specifically, a comparison between the proposed method and those methods used nowadays to fabricate superhydrophobic surfaces in terms of the durability of the super- hydrophobic surface and the cost and the production efficiency of the method.
Microstructures prepared by laser have strong adhesion to substrates and are more durable than most of methods used nowadays to fabricate superhydrophobic surfaces. Compared with Femtosecond and picosecond lasers, the processing efficiency of infrared laser is much higher. For example, femtosecond laser takes three hours to fabricate a 20*20*2mm superhydrophobic sample, whereas infrared laser takes only 15 minutes. The cost of a femtosecond laser is in the millions, while an infrared laser is in the hundreds of thousands. In a word, the superhydrophobic surface prepared by infrared laser has the characteristics of low cost, high efficiency and high durability.
Is there a linear relationship between the contact angle and the time interval between measures?
The contact angle decreased slowly with the increase of time. However, the decrease is very small and the surface still exhibits super-hydrophobic property.
A reference should be added too or a more detailed explanation in order to explain what is considered “good durability” (line 222).
We have added a reference to explain what is considered “good durability” (line 220).
In line 12, the format of the units “mm s-1” should be revised.
We have revised typo errors.
In line 48, and according with the previous paragraph a reference or a more detailed explanation should be added in order to clarify the sentence “Inexpensive and stable super-hydrophobic surfaces are easily fabricated”.
We have added a reference to clarify the sentence.
In line 53, the word “para-meters” should be revised.
We have revised the word.
In line 67, the number of the micro and nanometer-scale structures used in the analysis should be indicated as well as the deviation of the average values (line 80) and the description of the positions used to measure the samples (line 80). This could help to better understand the results show in figure 6.
The number of micro and nanometer-scale structures is 25-300μm. The five measured positions are near the four boundaries of the sample and at the center of the sample. The deviation of the average values is ±0.8 °.
In line 94, the format of the units “mm s-1” should be revised.
In line 100 “is10W” should be revised.
In line 107-109, figure 2(b) is associated whit a scanning speed of 1000mm/s but in line 101 Figure 2(b) indicates 500mm/s.
In line 161-166, the format of the paragraph is different from that in the rest of the paper.
We have revised all of these typo errors.
Figure 1, 2, 3, 5, the indicators (a), (b), (c) and (d) are in white inside the image and are hard to see. The quality of Figure 9 should be improved.
We re-edited these Figures.
I greatly appreciate your help to this paper. I hope that the revised manuscript is now suitable for publication.
Thank you

Reviewer 2 Report
This manuscript describes a facile method of fabricating superhydrophobic surfaces on Al alloy by using an infrared laser textured and chemically modified method. The authors carried out a detailed investigation on varying many parameters on the fabrication of Al-based micro-nanostructures. The manuscript is interesting and worthy of publication in Materials. Few minor comments should be addressed before publication.
Few typo errors. For example: Line: 236: “The crater micro-structure was textured on the surface of simple”. Line: 20 “potential of wide ranging applications”. Include the applications. What is the composition of Al 6061 alloys? Did authors observe any change in composition of Al 6061 alloys after laser treatment? Why specifically infrared laser?Author Response
Dear reviewer:
Many thanks for the insightful comments and suggestions of you. I have made corresponding revision according to your advice. Words in red are the changes I have made in the text.
The following is the answers and revisions I have made in response to your questions and suggestions.
Few typo errors.
Typo errors have been corrected.
What is the composition of Al 6061 alloys?
The composition of Al 6061 alloys is as follow: Mg: 0.8-1.2%, Si: 0.4-0.8%, Fe: 0.7%, Cu: 0.15-0.4%, Mn: 0.15%, Cr: 0.04-0.35%, Zn: 0.25%, Ti: 0.15%. The rest is Al.
Did authors observe any change in composition of Al 6061 alloys after laser treatment?
We observed some changes in composition of Al 6061 alloys after laser treatment, such as: the increase of Oxygen element. But we haven't studied the mechanism systematically. This idea will be one of our future research directions.
Why specifically infrared laser?
Because of low cost and high processing efficiency, infrared laser has great advantage in mass production of microstructure surface. For example, femtosecond laser takes three hours to fabricate a 20 mm *20 mm *2mm superhydrophobic sample, whereas infrared laser takes only 15 minutes. The infrared laser has not been used in the mass production because of the limitation of the preparation technology. Therefore, the study of the influence of the infrared laser parameters on the hydrophobic performance will promote the application of the super-hydrophobic surface.
I greatly appreciate your help to this paper. I hope that the revised manuscript is now suitable for publication.
Thank you

Reviewer 3 Report
The manuscript "Infrared laser textured aluminum alloy surfaces with stable superhydrophobicity" presents infrared laser scanning process coupled with a simple salinization modification to improve the hydrophobicity of aluminium alloy surface.
The main issue of this manuscript is the novelty as similiar research been made before with better characterizations, not cited in this script.
L. B. Boinovich, A. M. Emelyanenko, A. D. Modestov, A. G. Domantovsky,
and K. A. Emelyanenko, Synergistic Effect of Superhydrophobicity and Oxidized Layers on Corrosion Resistance of Aluminum Alloy Surface Textured by
Nanosecond Laser Treatment, ACS Appl. Mater. Interfaces 2015, 7, 19500−19508
J. T. Cardoso A. Garcia-Giron,J. M. Romano, D. Huerta-Murillo, R. Jagdheesh, M. Walker, S. S. Dimov and J. L. Ocana, Influence of ambient conditions on the evolution of wettability properties of an IR-, ns-laser textured aluminium alloy, RSC Adv., 2017, 7, 39617
The other issues are the limited methods with only shown SEM and contact angle measurements.
To bring this manuscript in a publishable form the authors have to give in introduction why their methods better than previous published one and where is their novelty. Additional experiment as EDX and electrochemical methods such as cyclic voltammetry and impedance measurements will give more inside of the properties of their laser induced surface change.
Another point is the missing discussion in the result part, which has to be included.
The SEM images need to be replaced with a better image as they difficult to distinguish.
Also the title differ from submission (Study of Infrared Laser Parameters on Surface Morphology and Hydrophobic Properties) and script (Infrared laser textured aluminum alloy surfaces with stable superhydrophobicity). Please correct that.
Author Response
Dear reviewer:
Many thanks for the insightful comments and suggestions of you. I have made corresponding revision according to your advice. Words in red are the changes I have made in the text.
The following is the answers and revisions I have made in response to your questions and suggestions.
The other issues are the limited methods with only shown SEM and contact angle measurements. To bring this manuscript in a publishable form the authors have to give in introduction why their methods better than previous published one and where is their novelty. Additional experiment as EDX and electrochemical methods such as cyclic voltammetry and impedance measurements will give more inside of the properties of their laser induced surface change.
We want to find out the ideal range of infrared laser parameters, so as to provide a reference for mass production of superhydrophobic surface in the future. Thus, we mainly discuss the difference of surface microstructure morphology under different laser parameters and the influence of changing the microstructure morphology on the surface wettability. Therefore, we just show SEM and contact angle measurements. Our follow-up work will focus on the characterization of surface properties.
Another point is the missing discussion in the result part, which has to be included.
We add some discussion in the result part.
The SEM images need to be replaced with a better image as they difficult to distinguish.
We re-edited these Figures.
Also the title differ from submission (Study of Infrared Laser Parameters on Surface Morphology and Hydrophobic Properties) and script (Infrared laser textured aluminum alloy surfaces with stable superhydrophobicity). Please correct that.
The title (Study of Infrared Laser Parameters on Surface Morphology and Hydrophobic Properties) is wrong, we use this one (Infrared laser textured aluminum alloy surfaces with stable superhydrophobicity).
I greatly appreciate your help to this paper. I hope that the revised manuscript is now suitable for publication.
Thank you

Round 2
Reviewer 3 Report
All open question and concerns reasonable answered. Manuscript can be published in its form